# What Will My Model Forget? Forecasting Forgotten examples in Language Model Refinement

## Abstract

Language models deployed in the wild make errors. However, simply updating the model with the corrected error instances causes catastrophic forgetting—the updated model makes errors on instances learned during the instruction tuning or upstream training phase. Randomly replaying upstream data yields unsatisfactory performance and often comes with high variance and poor controllability. Precisely identifying forgotten examples is computationally intractable with a large upstream dataset. To this end, we study the problem of forecasting upstream examples that will be forgotten due to a model update. We shed light on how interactions between examples cause learning one example to forget the other. We train forecasting models given a collection of online learned examples and corresponding forgotten upstream pre-training examples. We propose a partially interpretable forecasting model based on the observation that changes in pre-softmax logit scores of pretraining examples resemble that of online learned examples, which performs decently on BART but fails on T5 models. We further show a black-box classifier based on inner products of example representations achieves better forecasting performance over a series of setups. Finally, we show that we reduce forgetting of upstream pretraining examples by replaying examples that are forecasted to be forgotten, demonstrating the practical utility of forecasting example forgetting across different setups.

## 1 Introduction

While pretrained language models (PTLMs) have achieved remarkable success in various downstream tasks, it is inevitable that models deployed in the wild still make errors (Lin et al., 2022b; OpenAI, 2023). Fixing errors without retraining the model, known as model refinement (Yao et al., 2021), is crucial for the long-term usability of the model (Raffel, 2023). Although a few steps of parameter updates are usually sufficient to correct errors (De Cao et al., 2021), a main obstacle is catastrophic forgetting, *i.e.,* massive misprediction of previously learned examples (Robins, 1995). To combat forgetting, a prevalent practice in model refinement or continual learning algorithms is to replay previously learned examples, most of which rely on random sampling from upstream training data (de Masson D'Autume et al., 2019; Jin et al., 2022). However, such practice has shortcomings that (1) they lack interpretability on what previously learned examples are affected due to model updates, and (2) they achieve inferior performance as the replayed examples are not properly targeted.

Few works have tried to analyze or forecast forgotten examples in model updates. Existing work demonstrated the existence of examples that are more prone to forgetting (Toneva et al., 2018; Tirumala et al., 2022; Maini et al., 2022); however, they do not interpret how interactions between two examples contribute to forgetting, *i.e.*, why learning one example causes forgetting of the other. For PTLMs, such interaction is intriguing to humans, as exemplified in Figure 1, where learning an example about public relations causes forgetting of an example in paraphrase detection. This opens up a novel and challenging task of forecasting forgetting based on interactions of two examples without running expensive and repetitive inference with PTLMs.

The goals of this work are two fold: (1) shedding light on how interactions between two examples contribute to forgetting, and (2) developing effective methods that forecast example forgetting.

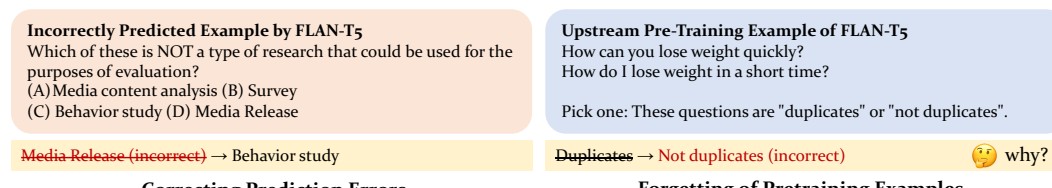

Figure 1: Intriguing patterns of example forgetting while correcting prediction errors in FLAN-T5. Fixing errors in a question related to public relations flip the prediction on an example from the paraphrase detection task. The association between two examples is obscure to humans.

Towards the goal of interpretability, we explore forecasting forgetting with an interpretable model. With empirical study, we demonstrate the phenomenon of "logit-change transfer", *i.e.*, the changes in pre-softmax logits of upstream pretraining examples proportionally copy that of online learned examples while fixing an error, causing forgetting of the upstream pretraining example. Motivated by this finding, we build a partially interpretable forecasting model that learns how much logit changes are transferred based on the similarity of two examples. Similar techniques have been applied to track the dynamics of logits during continual learning in recent works (Ramasesh et al., 2020; Karakida & Akaho, 2021; Evron et al., 2022). Experiments show that the forecasting model is effective on BART0 (Lin et al., 2022a; Lewis et al., 2019) but fails on FLAN-T5 (Chung et al., 2022).We then examine whether a black-box model can achieve better forecasting performance than the interpretable model. We show that a model based on inner products of trainable representations of two examples could achieve decent forecast accuracy across a series of setups.

We further demonstrate the practical utility of forecasting forgotten examples. At inference time, the forecasting model is highly computationally efficient and does not require inference with PTLMs. By replaying examples predicted to be forgotten, we reduce catastrophic forgetting compared to replaying random examples. The approach is a significantly efficient alternative to replaying exact examples that will be forgotten (Aljundi et al., 2019).

To summarize, our contributions are three fold: (1) a novel problem setup of forecasting forgotten examples in model refinement, (2) a partially interpretable and a black-box model for forecasting forgetting, and (3) a model refinement algorithm with reduced forgetting by replaying examples predicted to be forgotten.

## 2 FORECASTING FORGOTTEN EXAMPLES

We set up a formal problem formulation of forecasting examples that will be forgotten in model refinement. We assume that a base language model (LM), $f_0$, is pretrained on a collection of upstream data $D_{\text{PT}}$. We also assume $f_0$ to be an instruction-tuned model (Chung et al., 2022) that can perform diverse natural language tasks in $D_{\text{PT}}$ in a format of sequence-to-sequence generation. We measure Exact Match (EM) score of a model $f$ on a dataset $D$, defined as $\text{EM}_{D,f} := |\{\langle x, y \rangle \in D \mid f(x) = y\}| / |D|$, where $x$ is the input text and $y$ is the ground truth answer.

**Model Refinement.** We evaluate the LM $f_0$ a new task and collect all the mispredicted examples, noted as $D_{\text{R}}$. For each $\langle x_i, y_i \rangle \in D_{\text{R}}$, we fine-tune the language model $f_0$ for $K$ steps and obtain an updated model $f_i$ for each of $\langle x_i, y_i \rangle$. We evaluate **Edit Success Rate**, defined as $|\{\langle x_i, y_i \rangle \in D_{\text{R}} \mid f_i(x_i) = y_i\}| / |D_{\text{R}}|$, *i.e.*, the proportion of examples that produces correct answers after model updates. After fine-tuning on each of $\langle x_i, y_i \rangle$, we measure **EM Drop Ratio** on $D_{\text{PT}}$, defined as $(\text{EM}_{D_{\text{PT}}, f_i} - \text{EM}_{D_{\text{PT}}, f_0}) / \text{EM}_{D_{\text{PT}}, f_0}$. In our default setup, we only consider fixing one error at a time; we involve sequential model updates over multiple examples later in our experiments.

**Forecasting Forgetting**. Due to forgetting, among the subset of upstream pretraining examples in $D_{\text{PT}}$ that are correctly classified by the base pretrained LM $f_0$, $\hat{D}_{\text{PT}} := \{\langle x_j, y_j \rangle \in D_{\text{PT}} \mid f_0(x_j) = y_j\}$, examples may get correct or incorrect predictions after updating $f_0$ on $\langle x_i, y_i \rangle$. For each online learned example $\langle x_i, y_i \rangle$, we collect forgotten upstream pretraining examples, $D_{\text{PT}}^{\text{Fgt},i} := \{\langle x_j, y_j \rangle \in \hat{D}_{\text{PT}} \mid f_i(x_j) \neq y_j\}$ and those not forgotten, $D_{\text{PT}}^{\text{Non-Fgt},i} = \hat{D}_{\text{PT}} \backslash D_{\text{PT}}^{\text{Fgt},i}$. The task of forecasting forgetting is a binary classification problem $g : \langle x_i, y_i \rangle, \langle x_j, y_j \rangle \mapsto z_{ij} \in \{0, 1\}$ where the posi-

tive class corresponds to $\langle x_j, y_j \rangle$ being forgotten upon learning $\langle x_i, y_i \rangle$. We note that although the ground truth of forgotten upstream pretraining examples can be directly obtained by running inference on $\hat{D}_{\text{PT}}$ with the updated model $f_i$, it can be very inefficient and repetitive assuming a large $\hat{D}_{\text{PT}}$ and $\hat{D}_{\text{R}}$. We expect the forecasting function $g$ to be computationally efficient, and prohibit $g$ from running full inference over $\hat{D}_{\text{PT}}$ repetitively for each online learned example $\langle x_i, y_i \rangle \in D_{\text{R}}$.

**Training and Evaluation of Forecasting Methods.** We partition the set of online learned examples into disjoint subsets $D_{\text{R}}^{\text{Train}}$ and $D_{\text{R}}^{\text{Test}}$. Given a trainable forecasting model $g$ of example forgetting, we train $g$ using $\langle D_{\text{R}}^{\text{Train}}, D_{\text{PT}} \rangle$ and evaluate it with $\langle D_{\text{R}}^{\text{Test}}, D_{\text{PT}} \rangle$. We also evaluate out-of-domain generalization performance of $g$ where $D_{\text{R}}^{\text{Train}}$ and $D_{\text{R}}^{\text{Test}}$ are from different tasks.

# 3 METHODS

The key to the challenge of forecasting forgetting is to develop a computationally efficient and reliable forecasting function $g$; Furthermore, a self-interpretable $g$ could assist humans to understand why an example is forgotten. In this section, we introduce two (partially) interpretable approaches for forecasting forgetting inspired by empirical findings about frequently forgotten examples and logit changes of examples. We also present a black-box forecasting model based on inner products of representations of two examples.

## 3.1 FREQUENCY-THRESHOLD BASED FORCASTING

Existing study by Toneva et al. (2018); Maini et al. (2022) shows the existence of examples that are more prone to forgetting than others. Following these findings, we set up a baseline that predicts positive if the example is forgotten more than a preset frequency threshold $\gamma$ in $D_{\text{R}}^{\text{Train}}$.

$$g(\langle x_i, y_i \rangle, \langle x_j, y_j \rangle) = \mathbb{1}[|\{\langle x_i, y_i \rangle \in D_{\text{R}}^{\text{train}} \mid z_{ij} = 1\}| \geq \gamma] \tag{1}$$

The threshold $\gamma$ is tuned to maximize its F1 on $D_{\text{R}}^{\text{train}}$. We refer to the approach as **threshold-based forecasting**. However, this baseline does not capture or interpret how interactions between the online learned example $\langle x_i, y_i \rangle$ and the pretraining example $\langle x_j, y_j \rangle$ contribute to the forgetting.

## 3.2 LOGIT-CHANGE BASED FORECASTING

As we have seen in Figure 1, it is intriguing to humans why learning an example $\langle x_i, y_i \rangle$ (about public relations) causes model to forgetting an upstream pretraining example $\langle x_j, y_j \rangle$ (about paraphrase detection). We figure out clues by examining logit change (pre-softmax outputs) of two examples after model updates. Figure 2(a) reveals that in the same pair of examples, the logit scores of the some tokens such as "not" and "duplicates" in $\langle x_i, y_i \rangle$ changes significantly, despite that their token *probabilities* after normalization are close to 0. This logit change does not have an effect on the prediction of $\langle x_i, y_i \rangle$; but the problem arises on $\langle x_j, y_j \rangle$ as the logit change partially transfers to the example. In the pretraining example $\langle x_j, y_j \rangle$, the logit change affects the ordering of the top-2 prediction candidates ("not" versus "duplicates"), causing the prediction to flip.

A natural question is whether we can predict the proportion of logit changes of $\langle x_i, y_i \rangle$ that will be transferred to $\langle x_j, y_j \rangle$. We derive the relationships between logit change of the online learned example and the pretraining example with techniques similar to those of previous work on neural tangent kernels (NTKs) (Lee et al., 2019). We note the output logits of an example $x$ as $\hat{f}(x) \in \mathbb{R}^{TV}$, where $T$ is the output length, and $V$ is the size of the vocabulary. The change of model parameters $\Delta\theta_i = \theta_i - \theta_0$ in the model $f$ after a single step of gradient step on the online learning example $\langle x_i, y_i \rangle$ is $\theta_i - \theta_0 = -\eta\nabla_\theta\hat{f}_0(x_i)\nabla_{\hat{f}_0(x_i)}\mathcal{L}(x_i, y_i)$, where $\mathcal{L}$ is the training loss function and $\eta$ is the learning rate. With the first-order Taylor expansion, the logit change of an upstream pretraining example $x_j$ after performing one step of gradient descent with $x_i$ can be approximated as $\Delta\hat{f}_i(x_j) = \hat{f}_i(x_j) - \hat{f}_0(x_j) = -\eta\Theta(x_j, x_i)\mathcal{L}(x_i, y_i)$, where the kernel $\Theta(x_j, x_i) \in \mathbb{R}^{TV \times TV}$ measures inner products among gradients $\nabla_\theta\hat{f}_0(x_j)\nabla_\theta\hat{f}_0(x_i)^T$. Similarly, for the online learning example, the logit change is $\Delta\hat{f}_i(x_i) = \hat{f}_i(x_i) - \hat{f}_0(x_i) = -\eta\Theta(x_i, x_i)\mathcal{L}(x_i, y_i)$. We therefore obtain the relationship between the logit changes of $x_i$ and $x_j$,

$$\hat{f}_i(x_j) - \hat{f}_0(x_j) = \Theta(x_j, x_i)\Theta^{-1}(x_i, x_i)[\hat{f}_i(x_i) - \hat{f}_0(x_i)] \tag{2}$$

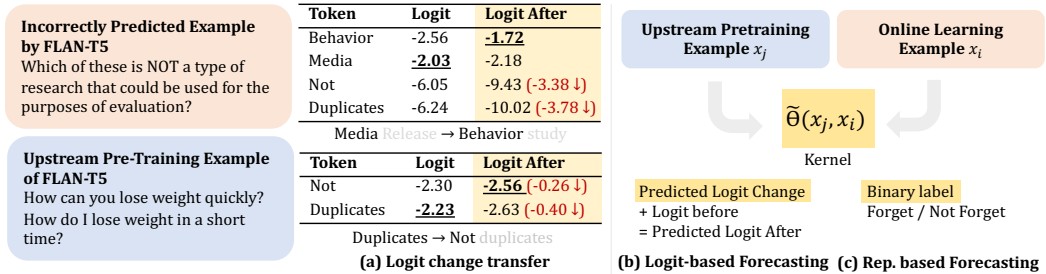

Figure 2: (a) **Transfer of logit changes** of first output tokens on an upstream pretraining example $\langle x_j, y_j \rangle$ when fixing prediction errors of an online learning example $\langle x_i, y_i \rangle$ (see Figure 1 for the full texts of the example). After fixing the error, the logit scores of the tokens "not" and "duplicates" in $\langle x_i, y_i \rangle$ changes significantly, despite that their token *probabilities* after normalization are both close to 0. The logit change has no effect on the prediction of $\langle x_i, y_i \rangle$; however, the predictions of the upstream pretraining example $\langle x_j, y_j \rangle$ flips as the logit change partially transfers to $\langle x_j, y_j \rangle$. (b) **Logit-based forecasting** infers transfer of logit changes depending on the learned similarity measurement of two examples. (c) **Representation-based forecasting** directly predicts the binary label of forgetting based on learned similarity measurement.

Eqn. 2 enables forecasting logit change of a pretraining example $x_j$ with the logit change of the online learned example $x_i$ and the kernel matrices $\Theta(x_j, x_i)\Theta^{-1}(x_i, x_i)$. Nevertheless, the kernel can be either easy or notoriously expensive to compute, depending on the trainable parts of the models. When only fine-tuning the LM heads, obtaining $\Theta(x_j, x_i)$ does not require running repetitive inference with the LM because the gradients $\nabla_{W_{\text{Head}}} \hat{f}_0(x_j)$ are simply the representations of $x_j$ before the LM head, which can be computed once and cached for each $x_j \in D_{\text{PT}}$. We refer to the approach as **fixed logit-based forecasting**. Unfortunately, when more parts of $f_0$ are fine-tuned, the kernel requires $TV$ backward passes to obtain the gradients, which is prohibitive (Novak et al., 2022).

**Trainable Logit-Based Forecasting Model.** Is it possible that for LMs, we can approximate the learning dynamics in Eqn. 2 with a low-dimensional and simpler model? We examine a significantly simplified alternative of Eqn. 2 by substituting $\Theta(x_j, x_i)\Theta^{-1}(x_i, x_i)$ with a trainable kernel $\tilde{\Theta}(x_j, x_i) = h(x_j)h(x_i)^T$, where $h : x \mapsto \mathbb{R}^{T \times d}$ is an encoding function that maps $x$ to a low-dimensional vector in $\mathbb{R}^d$, where $T$ is output length. We implement $h$ with a trainable LM and extract its representation in the final layer as $h(x)$. We also remove the huge $30k - 50k$ dimensional vocabulary space from the kernel so that $\tilde{\Theta}(x_j, x_i) \in \mathbb{R}^{T \times T}$. As such, we forecast the logits (reshaped from $\mathbb{R}^{TV}$ to $\mathbb{R}^{T \times V}$) of $x_j$ in the updated LM $f_i$ as $\hat{f}_i(x_j) = \tilde{\Theta}(x_j, x_i)[\hat{f}_i(x_i) - \hat{f}_0(x_i)] + \hat{f}_0(x_j)$. Note that the equation does not require repetitive inference with the LM $f$: the logits of pretraining examples before model updates $\hat{f}_0(x_j)$ can be computed once and cached for different online learning examples $x_i$; the same applies to the representations $h(x_i)$ required by the trainable kernel $\tilde{\Theta}$. Upon forecasting the logits of pretraining examples after model updates, we optimize a margin loss so that the predicted logit score $\hat{f}_i(x_j)[y_j]$ of the correct token $y_j$ exceeds the second-top candidate token $\max_{v \neq y_j} \hat{f}_i(x_j)[v]$ by a preset margin if $\langle x_j, y_j \rangle$ is not forgotten, and reversed otherwise.

$$\mathcal{L}(\langle x_i, y_i \rangle, \langle x_j, y_j \rangle, z_{ij}) = (1 - z_{ij}) \max(0, 1 - (\hat{f}_i(x_j)[y_j] - \max_{v \neq y_j} \hat{f}_i(x_j)[v]))$$
$$+ z_{ij} \max(0, 1 - (\max_{v \neq y_j} \hat{f}_i(x_j)[v] - \hat{f}_i(x_j)[y_j])) \tag{3}$$

We summarize the full training and evaluation procedure in Algorithms 1 and 2 in Appendix C. The resulting forecasting model is partially interpretable in that it explains how the logit change of $\langle x_i, y_i \rangle$ is transferred to $\langle x_j, y_j \rangle$ depending on their similarity. We illustrate the method in Figure 2(b). In our experiments, we notice that this greatly simplified logit-based forecasting is effective on BART models, but fails on another model, T5. The findings implies that logit change transfer cannot be captured by a simplified kernel $\tilde{\Theta}$ for all model types.

### 3.3 REPRESENTATION-BASED FORECASTING

We examine whether a black-box forecasting model can extract latent interactions between two examples $x_i$ and $x_j$ that contribute to forgetting. We propose a model that directly maps the inner

products of the representations $h(x_j)h(x_i)^T$ (*i.e.,*, the kernel $\tilde{\Theta}(x_j, x_i)$) to the label $z_{ij} \in \{0, 1\}$.

$$g(\langle x_i, y_i \rangle, \langle x_j, y_j \rangle) = \sigma(\sum_{t=1}^{T} h(x_{j,t}) \sum_{t=1}^{T} h(x_{i,t})^T) \tag{4}$$

where $T$ is the length of the output sentence and $h$ is a trainable LM encoder similar to trainable logit-based forecasting. We optimize binary cross-entropy loss to learn the encoding function $h$. We illustrate the method in Figure 2(c).

**Forecasting with Frequency Priors.** We expect the representation-based forecasting model to capture latent interactions that cannot be learned by the threshold-based forecasting in Sec. 3.1. However, it is likely that the model overfits such a bias about frequency of example forgetting. Therefore, we add a bias term to Eqn. 4 to represent the frequency priors before training, so that the model fits the residuals. The bias term is the log odds that the example is forgotten in $D_R^{\text{Train}}$, more specifically $b_j = \log(|\{\langle x_i, y_i \rangle \in D_R^{\text{train}} \mid z_{ij} = 1\}| \ / \ |D_R^{\text{train}}|) - \log(|\{\langle x_i, y_i \rangle \in D_R^{\text{train}} \mid z_{ij} = 0\}| \ / \ |D_R^{\text{train}}|)$. We refer to the approach as **representation-based forecasting**. We summarize the full training and inference procedures in Algorithms 3 and 4 in Appendix C. In our experiments, we will present the results of ablating the frequency prior.

## 4 EXPERIMENT SETUPS

Our main goals of experiments are (1) to examine the performance of methods that forecast forgetting when fixing errors in PTLMs and (2) to evaluate the practical utility of forecasting methods by replaying examples predicted to be forgotten. We experiment with multiple PTLMs, datasets, and various fine-tuning setups of PTLMs.

### 4.1 TRAINING AND EVALUATION SETUP

**Base PTLMs and Datasets ($f_0$, $D_{\text{PT}}$).** We experiment with BART0$_{\text{Large}}$ (Lewis et al., 2019; Lin et al., 2022a), FLAN-T5$_{\text{Large}}$, FLAN-T5$_{3B}$ (Chung et al., 2022) models with 400M, 840M, and 3B parameters respectively. All of these models are encoder-decoder language models instruction-tuned over a mixture of training tasks and are able to solve diverse tasks in a format of sequence-to-sequence generation. We evaluate forgetting over 36 tasks from the training split of the Public Pool of Prompts (P3) dataset (Bach et al., 2022), which is involved in pretraining all three base PTLMs.

**Tasks for Model Refinement ($D_{\text{R}}$).** We collect mispredicted examples (with Exact Match (EM) metrics) of PTLMs over datasets that are not involved for pretraining. For BART0, we use tasks from the test split of the P3 dataset; For FLAN-T5, we use MMLU (Hendrycks et al., 2020), since the P3 dataset (including the test split) is involved in pretraining the model.

**Training and Evaluation of the Forecasting Model $g$.** For a given dataset for model refinement, we collect mispredicted examples from the training split $D_R^{\text{Train}}$ and the validation split $D_R^{\text{Test}}$. We train the forecasting model with $D_R^{\text{Train}}$ and report the performance on $D_R^{\text{Test}}$. We then evaluate the performance of correcting these errors of PTLMs on $D_R^{\text{Test}}$. We fine-tune the entire model (Full FT), low-rank learnable weights (LoRA) (Hu et al., 2021), or the LM head only.

**Hyperparameters.** We perform 30 steps of parameter updates on a single online learning example to fix the error when we apply LoRA or full FT, and 100 steps when we only fine-tune the heads. For LoRA and full fine-tuning we use a learning rate of $10^{-5}$ for BART0$_{\text{Large}}$ and $10^{-4}$ for FLAN-T5. When fine-tuning heads only, we use learning rates of $10^{-3}$ or $10^{-4}$. The hyperparameters of model refinement are tuned to maximize edit success rate, which is the primary goal of model refinement. We leave other details in Appendix B.

**Metrics.** We report **F1** scores for binary forgetting prediction. For model refinement, we report the **Edit Success Rate** and **EM Drop Ratio** defined in Sec. 2.

### 4.2 COMPARED METHODS

**Forecasting Forgetting.** For forecasting forgetting, we report the performance of fixed logit-based, trainable logit-based and black-box representation-based forecasting. The fixed logit-based fore-

Table 1: Average F1-score of forecasting example forgetting when fixing one error in $D_{\mathrm{R}}^{\mathrm{Test}}$ at a time. When fixing errors of base PTLMs, we either fine-tune LM heads only (Head), learn low-rank parameter updates (LoRA), or fine-tune entire model parameters (Full FT). Forgotten examples are minority among all pretraining examples. **Bold** numbers indicate the forecasting method that achieves the best performance.

| Language Model ($\rightarrow$) Dataset $D_{\mathrm{R}}$ ($\rightarrow$) | BART0$_{\mathrm{Large}}$ P3-Test | | FLAN-T5$_{\mathrm{Large}}$ MMLU | | | FLAN-T5$_{\mathrm{3B}}$ MMLU | |
|---|---|---|---|---|---|---|---|
| Method ($\downarrow$)  LM Tuning Setup ($\rightarrow$) | Head | Full FT | Head | LoRA | Full FT | Head | LoRA |
| Threshold | 62.96 | 55.75 | 59.95 | 43.93 | 48.43 | 63.64 | 41.42 |
| Fixed Logit | 69.57 | 43.26 | 68.37 | 19.54 | 12.74 | 59.03 | 17.50 |
| Trainable Logit | 73.39 | 57.15 | 61.09 | 36.54 | 40.91 | 55.07 | 31.40 |
| Representation | **79.32** | **67.19** | **67.81** | **48.66** | **51.51** | 65.93 | **42.99** |
| w/o Prior | 77.92 | 66.53 | 67.21 | 47.11 | 50.38 | 63.98 | 41.60 |

casting takes input representations before the LM head as its input, but this formulation only applies scenarios where we only fine-tune the LM heads; we report performance in other fine-tuning setups only as reference. We also note that forgotten examples (positive class) are minorities in $D_{\mathrm{PT}}$. As we will see in Sec. 5.2 and Table 3, EM Drop Ratio (that represents ratio of forgotten examples) ranges between 0.03% and 8.0%. This skewed distribution imposes challenges to forecasting methods.

**Model Refinement.** We correct errors in models with vanilla fine-tuning or randomly replaying a subset of examples from $D_{\mathrm{PT}}$. We perform replay with a distillation loss against the outputs of the base PTLM (Buzzega et al., 2020). We then verify whether replaying examples predicted as forgotten reduces forgetting. We also compare with an upper bound that replays ground-truth forgotten examples, which is computationally expensive and infeasible in practice, equivalent to Maximally Interfered Retrieval (Aljundi et al., 2019) with the entire pretraining set as retrieval candidates. For all variants of replay, we sparsely replay a mini-batch of 8 examples every 10 training steps on BART0$_{\mathrm{Large}}$ and FLAN-T5$_{\mathrm{Large}}$, and 4 examples every 5 training steps on FLAN-T5$_{\mathrm{3B}}$.

## 5 RESULTS

### 5.1 FORECASTING MODEL FORGETTING WHEN FIXING SINGLE ERROR

We evaluate the performance of forecasting example forgetting while fixing one single error in PTLMs. We examine whether these approaches outperform threshold-based forecasting that relies solely on the frequency of forgetting while ignoring interactions between examples in $D_{\mathrm{PT}}$ and $D_{\mathrm{R}}$. Table 1 summarizes the results.

**Performance when Tuning LM Heads Only.** We notice that on both BART0$_{\mathrm{Large}}$ and FLAN-T5$_{\mathrm{Large}}$, representation-based forecasting achieves the highest F1 (79.32 and 67.81). Fixed logit-based forecasting performs competitively, achieving F1 scores of 69.57 and 68.37, despite the absence of learnable parts in the method. As we discussed in Sec. 3.2, when only LM heads are tuned, fixed logit-based forecasting (Eqn. 2) can approximate the ground truth logit change of the upstream pretraining example. Still, F1 is not perfect due to the nature of the first-order approximation in Eqn. 2 and that we perform more than 1 gradient step to correct errors. Introducing trainable parts to logit-based forecasting at the cost of inexact formulation does not further improve performance. On BART0 and P3-Test, representation-based forecasting outperforms fixed logit-based forecasting (79.32 and 69.57 F1); while on FLAN-T5 and MMLU, the performance is close within two approaches (67.81 and 68.37 F1).

**Performance with LoRA or Full Fine-Tuning.** When we apply LoRA or fine tune the entire model to fix the errors, we see that the fixed logit-based forecasting no longer performs decently. Trainable logit-based forecasting (57.15 F1) can improve performance and outperform threshold-based prediction (55.75 F1) on BART0$_{\mathrm{Large}}$ , but not on FLAN-T5 models. As we discussed in Sec. 3.2, the reason is likely that trainable logit-based forecasting has greatly simplified the dynamics of logits in Eqn. 2; while for FLAN-T5, such a simplified model cannot fit the ground truth dynamics. Despite the failure of logit-based forecasting, representation-based forecasting performs competitively. On FLAN-T5$_{\mathrm{Large}}$ and MMLU, it improves F1 by 4.73 and 3.08 compared to threshold-based forecasting when fixing errors with full FT or LoRA updates. On BART0, the improvement is more significant,

Table 3: Edit success rate (Succ.) and Exact Match Drop Ratio (EM Drop %) of model refinement while fixing one error in $D_R^{\text{Test}}$ at a time. Lower EM Drop % indicates reduced forgetting. **Bold** numbers indicate lowest forgetting achieved by methods other than utilizing ground truth forgetting (GT Forget), which is computationally inefficient in practice. See Appendix B for EM scores of LMs on upstream data before refinement.

| Language Model ($\rightarrow$) Dataset $D_R$ ($\rightarrow$) | BART0$_{\text{Large}}$ P3-Test | | FLAN-T5$_{\text{Large}}$ MMLU | | | | FLAN-T5$_{\text{3B}}$ MMLU | |
|---|---|---|---|---|---|---|---|---|
| LM Tuning Setup ($\rightarrow$) | Full FT | | LoRA | | Full FT | | LoRA | |
| Methods ($\downarrow$) | Succ. | EM Drop % | Succ. | EM Drop % | Succ. | EM Drop % | Succ. | EM Drop % |
| Vanilla FT | 98.0 | 8.045 | 95.7 | 0.099 | 95.7 | 0.149 | 97.5 | 0.030 |
| Replay | | | | | | | | |
| w/ Random | 97.8 | 3.938 | 95.7 | 0.105 | 95.7 | 0.068 | 97.5 | −0.018 |
| w/ Threshold | 97.0 | 2.649 | 95.7 | 0.100 | 95.7 | 0.024 | 97.5 | 0.001 |
| w/ Trainable Logit | 97.3 | 2.250 | 95.7 | 0.113 | 95.7 | 0.081 | 97.5 | 0.004 |
| w/ Representation | 97.3 | **2.191** | 95.7 | **0.079** | 95.7 | **−0.026** | 97.5 | **−0.020** |
| w/ GT Forget | 97.0 | 0.401 | 95.7 | 0.075 | 95.7 | −0.056 | 97.5 | −0.011 |

with 11.41 higher F1 than threshold-based forecasting. We also notice consistent performance drop without the frequency prior term in representation-based forecasting.

**Out-of-Domain Generalization of Forgetting Prediction.** We evaluate whether forecasting models can generalize to new out-of-domain (OOD) tasks upon training on a mixture of in-domain tasks. We further split P3-Test into two disjoint subsets of tasks (details in Appendix B) and train the forecasting model on P3-Test$_{\text{ID}}$ while evaluating on P3-Test$_{\text{OOD}}$. We notice that although all trainable approaches outperform threshold-based prediction in terms of in-domain performance, only representation-based forecasting with frequency prior can improve OOD performance, obtaining an OOD F1 of 49.73, compared to 46.24 F1 of the threshold-based forecasting approach.

Table 2: In-domain (ID) and out-of-domain (OOD) performance of forgetting forecasting methods on BART0. We split P3-Test into in-domain and out-of-domain tasks and report performance on both splits.

| Method / Split | P3-Test$_{\text{ID}}$ | P3-Test$_{\text{OOD}}$ |
|---|---|---|
| Threshold | 60.45 | 46.24 |
| Trainble Logit | 64.15 | 30.61 |
| Representation | **75.11** | **50.12** |
| w/o Prior | 74.19 | 34.85 |

## 5.2 IMPROVING MODEL REFINEMENT WITH FORGETTING PREDICTION

We demonstrate the practical utility of forecasting forgetting by showing reduced catastrophic foregetting by replaying examples predicted to be forgotten. Table 3 summarizes the results of model refinement where we apply no replay (Vanilla FT), replay random examples, or replay examples predicted to be forgotten by forecasting methods. All variants of methods replay an equal number of examples from $D_{\text{PT}}$ at a fixed interval, as we described in Sec. 4.2.

Table 3 shows that EM Drop differs significantly between the BART (8%) and FLAN-T5 models when we do not apply replay (0.10%, 0.15% and 0.03%). Replaying random examples reduces EM Drop except LoRA fine-tuning on FLAN-T5$_{\text{Large}}$. Unsurprisingly, replaying ground truth forgotten examples is very effective for reducing forgetting. The comparison of EM Drop between replaying with threshold, trainable logit, and representation-based forecasting is mostly aligned with their F1 in forecasting forgotten examples in Table 1. Compared to replaying random examples, representation-based forecasting reduces EM Drop consistently in four different setups. However, the improvement is minor on FLAN-T5 models given the low forgetting when even no replay is performed (Vanilla FT). We will see more significant forgetting and clearer improvement of replaying forgotten examples when we continually fix multiple errors in the upcoming section. We also note that differences in replayed examples do not affect Edit Success.

## 5.3 CONTINUAL MODEL REFINEMENT OVER MULTIPLE EXAMPLES

**Generalization of Forecasting Models in Continual Model Refinement.** We examine whether forecasting models can generalize to scenarios of continually fixing multiple errors. The challenge

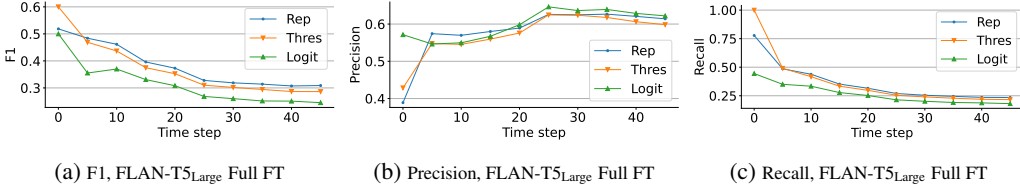

(a) F1, FLAN-T5$_{\text{Large}}$ Full FT    (b) Precision, FLAN-T5$_{\text{Large}}$ Full FT    (c) Recall, FLAN-T5$_{\text{Large}}$ Full FT

Figure 3: F1, Precision, and Recall of representation-based (Rep), threshold-based (Thres), and trainable logit-based forecasting models averaged up to a given time step (in $x$-axis) when continually refining the LM. For all forecasting methods, recall drops over time (as more examples being forgotten), while precision remains stable. Representation-based forecasting achieves best F1 and precision at the end of the sequence.

arises from the mismatch between the continually updated LM and the fixed pretrained LM ($f_0$) used for training forecasting models. Figure 3 plots the curves of averaged F1, Precision, Recall of forecasting up to a given time step while we continually fix the errors in the LMs. We notice that precision is mostly stable in the stream, while recall drops over time, mostly because more examples are forgotten over time. The stable precision indicates that the forecasting methods are effective in sequential model updates, but further improvement can be made to improve the recall of forecasting. The relative comparison between the forecasting methods also aligns with those of fixing single errors in Table 1, where the representation-based forecasting achieves the highest F1.

**Effect of Forecasting Forgetting in Continual Model Refinement.** We examine whether replaying examples predicted as forgotten during continual model updates reduces forgetting and summarize the results in Table 4. The results indicate much more significant forgetting of Vanilla FT compared to fixing single errors, where the EM drops are 5.5%, 3.3%, and 4.4% in three different setups of fine-tuning FLAN-T5 models. By replaying examples predicted as forgotten by the representation-based forecasting model, we significantly reduce the EM drops to 0.30%, 0.58%, and 0.14% respectively in three different setups.

Table 4: Exact Match Drop ratio (%) while continually fixing multiple errors. We report average scores of all examples in the stream.

| Model Dataset $D_R$ | FLAN-T5$_{\text{Large}}$ MMLU | | FLAN-T5$_{\text{3B}}$ MMLU |
|---|---|---|---|
| Tuning | LoRA | Full FT | LoRA |
| Vanilla FT | 5.463 | 3.302 | 4.384 |
| Replay | | | |
| w/ Random | 3.267 | 1.129 | 1.910 |
| w/ Threshold | 1.489 | 0.631 | 1.198 |
| w/ Trainable Logit | 2.565 | 0.898 | 1.516 |
| w/ Representation | **0.301** | **0.582** | **0.138** |
| w/ GT Forget | 0.189 | 0.560 | 0.030 |

## 5.4 DISCUSSION: COMPUTATIONAL EFFICIENCY

We discuss the computational efficiency of forecasting methods when retrieving forgotten examples from $N_{\text{PT}}$ upstream pretraining examples when fixing one error. We assume that the maximal lengths of the inputs and outputs are $T$, the feature dimensions of the sentence representations are $H$, and the size of the vocabulary is $V$. We denote the computational cost of running model inference with $N$ examples as $\text{Fw}(N)$ which can be very expensive given large LMs. We also consider that representations and logits of pretraining examples can be pre-computed, cached, and reused when forecasting forgetting caused by different online learning examples.

Table 5: Computational complexity of forecasting methods and obtaining ground truth forgetting by running inference with updated LMs when only fine-tuning the LM head or the entire model (Full FT). See Sec. 5.4 for the definitions of the notations.

| Method / Setup | Head | Full FT |
|---|---|---|
| Threshold | $O(N_{\text{PT}})$ | $O(N_{\text{PT}})$ |
| Trainable Logit | $O(N_{\text{PT}}T^2(H+V))$ | $O(N_{\text{PT}}T^2(H+V))$ |
| Representation | $O(N_{\text{PT}}H)$ | $O(N_{\text{PT}}H)$ |
| Ground Truth | $O(N_{\text{PT}}THV)$ | $O(\text{Fw}(N))$ |

Table 5 summarizes the computational efficiency of three forecasting methods and obtaining ground truth by running inference with updated LMs. The computational efficiency of forecasting approaches does not change when we only fine-tune LM heads or fine-tune the entire model. Obtaining the ground truth, in contrast, is less efficient when fine-tuning the entire LM compared to

when only LM heads are fine-tuned. Representation-based forecasting and threshold-based forecasting are more efficient than computing the ground truth in both setups. Logit-based forecasting is more efficient than computing ground truth when the maximal sequence length $T$ is small due to the term $T^2$ in its computational complexity. Nevertheless, all forecasting methods are far more efficient than computing the ground truth when the entire LM is fine-tuned because no repetitive inference with the LM is required. In Appendix D, we further present statistics about number of Floating Point Operations (FLOP), which aligns well with our computational complexity analysis.

## 6 RELATED WORKS

**Language Model Refinement**. Reserach on language model refinement studies efficient approaches to fix errors in LMs without retraining models from scratch (Yao et al., 2023). Several existing works focus on editing factual knowledge in LMs (Meng et al., 2022; Onoe et al., 2023; Zhang et al., 2023; Jang et al., 2021), while others, including this paper, study the problem in the context of general NLP tasks. De Cao et al. (2021); Mitchell et al. (2021) learn meta-models that edit update gradients to improve generalization of editing and reduce forgetting; Huang et al. (2023); Hartvigsen et al. (2022) add new neurons or adapters as patchers to fix errors. We note that our paper is focused on forecast forgetting, which brings the greatest benefit to replay-based model refinement and continual learning algorithms. Replay-based approaches are shown to perform competitively in various settings for PTLMs (de Masson D'Autume et al., 2019; Jin et al., 2022; Lin et al., 2022b; Wu et al., 2021). We compare with a non-replay model refinement algorithm by Mitchell et al. (2021) in Appendix A, which is effective in reducing forgetting, but at the cost of edit success rate in our setup.

**Empirical and Analytical Characterization of Example Forgetting.** Empirical study by Toneva et al. (2018) demonstrates that there exist examples that are more susceptible to forgetting. Maini et al. (2022) characterize training examples by their forgetting and inspect properties such as hardness or minority. This line of works on learning dynamics of single examples does not address interactions between examples that contribute to forgetting. Ramasesh et al. (2020) analytically study how learning new tasks may change affects logits of a learned task in a frozen feature model. Evron et al. (2022) analytically computes forgetting in linear models. Karakida & Akaho (2021) study learning dynamics in continual learning with neural tangent kernels (NTKs) (Jacot et al., 2018; Lee et al., 2019), and investigate conditions that cause knowledge transfer or forgetting between tasks. Unfortunately, NTKs are very expensive to compute for LMs with a large output space (Novak et al., 2022), which motivated us to approximate them with learnable models. In the context of large LMs, Tao et al. (2023) dissects the forgetting of encoders and classification heads by probing sentence representations given by LMs.

## 7 CONCLUSIONS

In this paper, we studied the problem of forecasting examples that will be forgotten when fixing errors in pretrained LMs. We set up problem formulation and evaluation protocols for forecasting example forgetting. We observe transfer of logit changes from an online learned example to an upstream pretraining example while fine-tuning PTLMs. Based on our empirical study on the logits of the upstream pretraining and online learning examples before and after model updates, we proposed a trainable logit-based forecasting method that infers the degree of logit change transfer. The approach performs well on BART0 but fails on FLAN-T5. We also proposed a black-box representation-based forecasting method that is consistently effective across various setups. We show that replay-based model refinement algorithms benefit from forecasting models and achieve reduced forgetting while fixing errors. Forecasting methods also generalize to sequential error fixing over multiple examples and reduce forgetting in the setup.

**Limitations.** Our experiments show that the success of logit-based forecasting methods depends on the type of model (which succeeds on BART0 but fails on FLAN-T5). Future work can analyze factors that affect the success of the approach and develop forecasting methods with similar level of interpretability but improved performance. Besides, although we showed that the performance of forecasting models generalizes from fixing single errors to sequentially fixing multiple errors, future works can study approaches to update forecasting models alongside the base pretraining model for more effective solutions in the setup of continual model refinement.

## REPRODUCIBILITY STATEMENT

All models (BART0, FLAN-T5) and datasets (P3, MMLU) used in the experiments in this paper are publicly accessible. We include training details during model refinement in Sec. 4.1 and introduce implementation details about forecasting methods in Appendix B. We will release the code upon acceptance of the paper.

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

## A    Comparison to non-replay model refinement methods

We briefly compare replay-based model refinement methods with MEND (Mitchell et al., 2021), which learns a meta model that edits gradients of model parameters when fixing the errors in LMs. We report the results in Table 6. We experiment with setup of LoRA fine-tuning only, because of the high cost of training meta models for the entire LM. We notice that EM Drop is even lower than replaying with ground truth forgotten examples, indicating the effectiveness of MEND in mitigating forgetting. However, we also notice that the edit success rate is lower than Vanilla FT and replay-based model refinement, which is the primary

Table 6: Comparison of edit succuss rate and EM Drop % between replay-based model refinement and MEND on FLAN-T5$_{\text{Large}}$ with LoRA fine-tuning.

|  | Edit Succ. | EM Drop% |
|---|---|---|
| Vanilla FT | 95.7 | 0.099 |
| Replay |  |  |
| w/ Random | 95.7 | 0.105 |
| w/ Representation | 95.7 | 0.079 |
| w/ GT Forget | 95.7 | 0.075 |
| MEND | 93.1 | 0.060 |
| w/o Forget Objective | 93.1 | 0.610 |

goal of performing model refinement in the first place. We also ablate the learning objective that mitigates forgetting, but we observe no improvement in the edit success rate.

## B    Implementation and Dataset Details

**Out-of-domain evaluation.** For out-of-domain evaluation of forecasting methods presented in Sec. 5.1, we partition P3-Test into two disjoint splits. We include SuperGlue-Cb, SuperGlue-RTE, SuperGLUE-wsc.fixed, SuperGlue-Copa, and SuperGlue-wic in the in-domain split, and include storycloze, hellaswag, anli, winograde-xl in the out-of-domain split.

**Training Details of the Forecasting Models.**
Both trainable logit-based and feature-based forecasting involve learnable encoders $h$ to encode input sentences. For BART0 experiments, we use BART0 followed by a freshly initialized 2-layer trainable MLP as the encoder $h$. For FLAN-T5 experiments, we use FLAN-T5$_{\text{small}}$ and a 2-layer MLP as the encoder. We optimize the LM components with a learning rate of $10^{-5}$, and the MLP with a learning rate of

Table 7: EM scores of base LM on the upstream training data (P3-Train) before performing updates.

| Model | EM |
|---|---|
| BART0$_{\text{Large}}$ | 50.50 |
| FLAN-T5$_{\text{Large}}$ | 47.47 |
| FLAN-T5$_{\text{3B}}$ | 51.31 |

$10^{-4}$. We train the forecasting models to a maximum of 100,000 steps with a batch size of 16. For each mini-batch, we sample 8 positive pairs ($\langle x_j, y_j \rangle$ is forgotten after learning on $\langle x_i, y_i \rangle$) and 8 negative pairs. During training, we assign a smaller weight ($\alpha$=0.1) to positive pairs due to the skewed nature of ground truth forgetting that occurs after updating the model, *i.e.*, the majority of examples are not forgotten and belong to the negative class.

**Base LM performance.** Table 7 summarizes the EM scores of the base LM on upstream data ($D_{\text{PT}}$) P3-train before performing updates. We note that BART0 is exclusively trained on P3-train, while FLAN-T5 models are trained on a mixture of other tasks with potentially different prompt formats. This interprets higher EM of BART0$_{\text{Large}}$ compared to FLAN-T5$_{\text{Large}}$.

## C    Details of Forecasting Algorithms

We summarize detailed procedures of training and inference of logit-based and representation-based forecasting methods in Algorithms 1, 2, 3, 4.

---

**Algorithm 1:** Training the logit-based forecasting model

---

**Data:** Training split of online learned examples $D_\text{R}^\text{train}$, upstream pretraining examples $D_\text{PT}$, Pretrained LM $f_0$, maximum input sentence length $T$

**Result:** Learned encoding function $h : \mathbb{R}^T \to \mathbb{R}^{T \times H}$

**while** *h has not converged* **do**

 Online learning example $\langle x_i, y_i \rangle \leftarrow$ sample($D_\text{R}^\text{train}$); Pretraining example $\langle x_j, y_j \rangle \leftarrow$ sample($D_\text{PT}$)

 Obtain logits $\tilde{f}_0(x_i)$ and $\tilde{f}_0(x_j)$

 $f_i \leftarrow$ update $f_0$ with $\langle x_i, y_i \rangle$

 Obtain updated logits $\tilde{f}_i(x_i)$ and $\tilde{f}_i(x_j)$

 Ground truth forgetting $z_{ij} \leftarrow 1$ if $f_0(x_i) \neq f_i(x_i)$ else $0$

 Encode $x_i$ to $h(x_i)$ and $x_j$ to $h(x_j)$;

 Compute the kernel matrix $\tilde{\Theta}(x_j, x_i) \in R^{T \times T} \leftarrow h(x_j)h(x_i)^T$

 Predict updated logits of $x_j$ as $\hat{f}_i(x_j) \leftarrow \tilde{\Theta}(x_j, x_i)[\hat{f}_i(x_i) - \hat{f}_0(x_i)] + \hat{f}_0(x_j)$

 Compute loss $\mathcal{L}(\langle x_i, y_i \rangle, \langle x_j, y_j \rangle, z_{ij})$ with Eq. 3 and optimize $h$

**end**

---

**Algorithm 2:** Inference with the trainable logit-based forecasting model

---

**Data:** Online learning example $\langle x_i, y_i \rangle \in D_\text{R}^\text{Test}$, upstream pretraining examples $D_\text{PT}$, Pretrained LM $f_0$, trained encoding function $h$, maximum input sentence length $T$, cached $h(x_j)$ for $\langle x_j, y_j \rangle \in D_\text{PT}$

**Result:** Predicted binary forgetting label $\hat{z}_{ij}$ on $D_\text{PT}$ for $\langle x_j, y_j \rangle \in D_\text{PT}$

Encode $x_i$ to $h(x_i)$

Obtain logits $\tilde{f}_0(x_i)$

$f_i \leftarrow$ update $f_0$ with $\langle x_i, y_i \rangle$

Obtain updated logits $\tilde{f}_i(x_i)$

**for** $\langle x_j, y_j \rangle \in D_\text{PT}$ **do**

 Encode $x_j$ to $h(x_j)$

 Compute the kernel matrix $\tilde{\Theta}(x_j, x_i) \in R^{T \times T} \leftarrow h(x_j)h(x_i)^T$

 Predict updated logits of $x_j$ as $\hat{f}_i(x_j) \leftarrow \tilde{\Theta}(x_j, x_i)[\hat{f}_i(x_i) - \hat{f}_0(x_i)] + \hat{f}_0(x_j)$

 **if** $\arg\max \hat{f}_i(x_j) \neq y_j$ **then**

  $\hat{z}_{ij} \leftarrow 1$

 **else**

  $\hat{z}_{ij} \leftarrow 0$

 **end**

**end**

---

**Algorithm 3:** Training the representation-based forecasting model

---

**Data:** Training split of online learned examples $D_\text{R}^\text{train}$, upstream pretraining examples $D_\text{PT}$, Pretrained LM $f_0$, maximum input sentence length $T$

**Result:** Learned encoding function $h : \mathbb{R}^T \to \mathbb{R}^{T \times H}$

**while** *h has not converged* **do**

 Online learning example $\langle x_i, y_i \rangle \leftarrow$ sample($D_\text{R}^\text{train}$); Pretraining example $\langle x_j, y_j \rangle \leftarrow$ sample($D_\text{PT}$)

 $f_i \leftarrow$ update $f_0$ with $\langle x_i, y_i \rangle$

 Ground truth forgetting $z_{ij} \leftarrow 1$ if $f_0(x_i) \neq f_i(x_i)$ else $0$

 Encode $x_i$ to $h(x_i)$ and $x_j$ to $h(x_j)$

 Obtaining the frequency prior

 $b_j \leftarrow \log(|\{\langle x_i, y_i \rangle \in D_\text{R}^\text{train} \mid z_{ij} = 1\}| / |D_\text{R}^\text{train}|) - \log(|\{\langle x_i, y_i \rangle \in D_\text{R}^\text{train} \mid z_{ij} = 0\}| / |D_\text{R}^\text{train}|)$

 Compute the probability of forgetting $\langle x_j, y_j \rangle$ as $\tilde{z}_{ij} \leftarrow \sigma(\sum_{t=1}^T h(x_{j,t}) \sum_{t=1}^T h(x_{i,t})^T) + b_j$

 Compute binary cross entropy loss $\mathcal{L}_\text{BCE}(\tilde{z}_{ij}, z_{ij})$ and update $h$

**end**

---

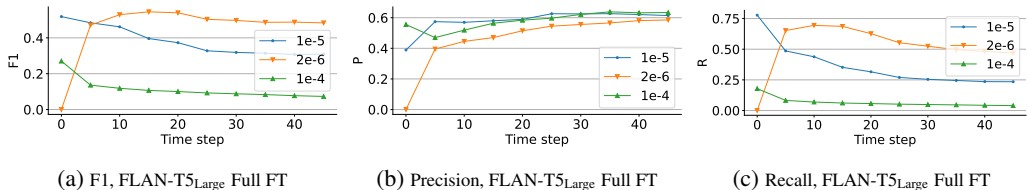

(a) F1, FLAN-T5$_{\text{Large}}$ Full FT  (b) Precision, FLAN-T5$_{\text{Large}}$ Full FT  (c) Recall, FLAN-T5$_{\text{Large}}$ Full FT

Figure 4: F1, Precision, and Recall of representation-based forecasting models averaged up to a given time step (in $x$-axis) when continually refining the LM under different learning rates.

---

**Algorithm 4:** Inference with the representation-based forecasting model

**Data:** Online learning example $\langle x_i, y_i \rangle \in D_R^{\text{Test}}$, upstream pretraining examples $D_{\text{PT}}$, Pretrained LM $f_0$, trained encoding function $h$, maximum input sentence length $T$, cached $h(x_j)$ for $\langle x_j, y_j \rangle \in D_{\text{PT}}$, cached frequency priors $b_j$ for $\langle x_j, y_j \rangle \in D_{\text{PT}}$
**Result:** Predicted binary forgetting label $\hat{z}_{ij}$ on $D_{\text{PT}}$ for $\langle x_j, y_j \rangle \in D_{\text{PT}}$
Encode $x_i$ to $h(x_i)$
**for** $\langle x_j, y_j \rangle \in D_{PT}$ **do**
  Encode $x_j$ to $h(x_j)$
  Compute the probability of forgetting $\langle x_j, y_j \rangle$ as $\tilde{z}_{ij} \leftarrow \sigma(\sum_{t=1}^{T} h(x_{j,t}) \sum_{t=1}^{T} h(x_{i,t})^T) + b_j$
**end**

---

## D   FLOATING POINT OPERATION COUNTS OF FORECASTING METHODS

We complement the computational complexity of forecasting methods with floating point operation (FLOP) statistics obtained during the experiments. We sample 100 examples per upstream task (36 tasks in total) to compute the statistics. Table 8 summarizes the results as we forecast forgetting when we update the model with a single online learning example. We see that representation-based and trainable logit-based forecasting require 1/6700 and 1/42 of FLOPs compared to obtaining ground truth forgetting by running inference on all upstream examples.

Table 8: Number of FLOPs when forecasting forgotten examples among 3,600 upstream pretraining examples given one online learning example.

| Method | #. FLOP |
|---|---|
| Representation | $1.35e^{10}$ |
| Trainable Logit | $2.15e^{11}$ |
| Ground Truth | $9.04e^{14}$ |

## E   HYPERPARAMETER ANALYSIS

### E.1   LEARNING RATES IN MODEL REFINEMENT

Learning rate is a crucial factor that trades off plasticity and stability during model updates. Table 9 shows that using a larger rate ($1e^{-4}$) than our default setup ($1e^{-5}$) clearly increases EM Drop ratio; while using a smaller learning rate ($2e^{-6}$) reduces EM Drop ratio at a cost of edit success rate. We further evaluate the forecasting model trained in the default setup on other learning rate setups and present the results in Figure 4. We notice that the precision scores almost remain the same across different learning rates, as a common subset of examples are forgotten; while recall scores differ across setups, because a greater number of examples are forgotten only when using larger learning rates. The results imply the precision scores of forecasting methods generalize well across different learning rate setups.

Table 9: Edit sccuess rate and EM Drop Ratio (%) under different learning rates in continual model refinement over multiple examples (Sec. 5.3) with Full FT on FLAN-T5$_{\text{Large}}$. $1e^{-5}$ is our default learning rate.

| Method | Succ. | EM Drop % |
|---|---|---|
| $1e^{-4}$ | 95.7 | 24.897 |
| $1e^{-5}$ | 95.7 | 3.302 |
| $2e^{-6}$ | 93.5 | 1.820 |

E.2    NUMBER OF REPLAYED EXAMPLES

As we introduced in Sec. 4.2, we replay a mini-batch of 8 examples every 10 update steps. This corresponds to replaying 3 mini-batches over 30 steps of model updates on a single online learning example. We also present the result of increasing the number of replayed examples by reducing intervals between replays while learning an online learning example. Table 10 summarizes the results. When fixing single errors, we notice that increasing the number of replayed examples causes increased forgetting (EM Drop Ratio). This is not surprising given previous studies that show over-fitting of models to replayed examples (Jin et al., 2022). Meanwhile, increasing the number of replayed examples consistently reduces the EM drop ratio when continually fixing multiple errors. By comparing to the results in Table 4, we see that the EM Drop Ratio of replaying 3 mini-batches of examples forecasted to be forgotten is between that of replaying 6 to 15 mini-batches of random examples.

Table 10: EM Drop Ratio (%) when replaying random example while fixing (1) single errors or (2) continually fixing multiple errors, which correspond to our setups in Tables 3 and 4 respectively. Replaying 3 mini-batches (one per 10 steps over 30 steps) corresponds to our default setup.

| #. replayed batches | Single errors | Multiple errors |
|---|---|---|
| 3 | 0.068 | 1.129 |
| 6 | 0.064 | 0.089 |
| 15 | 0.122 | 0.038 |
| 30 | 0.138 | -0.141 |

