# OpenReview forum: "What Will My Model Forget? Forecasting Forgotten Examples in Language Model Refinement"
_ICLR.cc/2024/Conference — Submitted to ICLR 2024_

### Official Review · Reviewer_PJD2 · 2023-10-14

**Soundness:** 4 excellent
**Presentation:** 4 excellent
**Contribution:** 3 good
**Rating:** 8
**Confidence:** 3

**Summary:**

The paper presents a method for mitigating catastrophic forgetting in instruction tuned language models, by building an interpretable forecasting model that predicts which training examples will be forgotten during model refinement, and replaying these examples. The authors compare a logit-based approach, which predicts how token logits on a pretraining example change before and after the model sees the online learning example, and a representation-based approach, which directly predicts whether the online learning example will cause a pretraining example to become erroneous using the latent representations of each example. Although the logit-based approach is more directly interpretable, the representation-based approach is robustly stronger at forecasting as well as mitigating forgetting across various settings (tuning heads only vs. LORA vs. all weights, different types of models).

**Strengths:**

This presents a novel and simple approach to tackle catastrophic forgetting, with clear problem formulation and strong motivation (Fig. 1-2 are very good). The paper is easy to read and clearly presented. The authors perform very thorough and thoughtful experiments that convincingly support their proposed method.

**Weaknesses:**

The computational efficiency discussion is very good, but I think reporting actual runtimes for some example settings would also be helpful. E.g. tuning the head only should be faster than the whole model, so I think the asymptotic complexity only tells half the story. It will also help practitioners consider how feasible this method is for their own application.

**Questions:**

Not sure if I missed this but why do the authors hypothesize the logit-based approach is less effective on FLAN vs BART?

---

> ### Author Response · Authors · 2023-11-20
> **Response to reviewer PJD2**
>
> We thank the reviewer for thoughtful and positive comments.
>
> **Q: Reporting actual runtimes for some example settings would also be helpful. E.g. tuning the head only should be faster than the whole model, so I think the asymptotic complexity only tells half the story.**
>
> We thank the reviewer for the suggestion. We agree that tuning the head is faster than tuning the whole model, and it is also more efficient to obtain ground truth forgotten examples when tuning the head only (assuming that pre-head outputs of upstream training examples can be cached). We discussed computation complexity of two cases separately in our initial version of Sec. 5.4, and concluded that our forecasting methods have better advantages when tuning the entire model.
>
> We additionally obtained the number of FLOP (number of floating point operations) of forecasting methods as an empirical statistic that measures computational efficiency, which is agnostic to the hardware that we use. We experimented with FLAN-T5-large with 100 examples sampled from each of the 36 pre-training tasks while fine-tuning the entire model.
>
> Method | FLOP
> -- | --
> Representation-based | $1.35e^{10}$
> Trainable Logit-based | $2.15e^{11}$
>
> As a comparison, the FLOP of running inference on a single example is $2.51e^{11}$ flops. Without forecasting methods, obtaining ground truth forgotten examples requires about $9.02e^{14}$ flops. Our forecasting methods are much more efficient.
>
> We have included the results in Appendix D and added pointers where we originally discussed computation complexity in Sec. 5.4.
>
> **Q: Why do the authors hypothesize the logit-based approach is less effective on FLAN vs BART?**
>
> We appreciate the question from the reviewer. In paragraph “Performance with LoRA or Full Fine-Tuning” in Sec. 5.1, we hypothesized that trainable logit-based forecasting has greatly simplified the dynamics of logits in Eqn. 2; for FLAN-T5, such a simplified model cannot fit the ground truth dynamics; in contrast, the learning dynamics of BART0 may be closer to a linear model and can be captured by our forecasting model. Empirically, we notice that FLAN-T5 models of all sizes (small, base, large, xl) present a similar behavior that logit-based forecasting is less effective. Therefore, we hypothesize that model size is not a key factor; instead, some details during the pre-training phase of these LMs may be crucial. We consider that dissecting factors that contribute to dynamics of logits during continual learning would be an interesting future work.

---

> > ### Comment · Reviewer_PJD2 · 2023-11-21
> > **Thanks authors**
> >
> > Thanks for the helpful response. It seems the main criticism of other reviews is that the empirical performance is not strong enough for the method to be practical. I believe the novelty of this method and the general performance improvements over baselines are sufficient to make this an interesting paper at ICLR, so I keep my accept rating.

---

### Official Review · Reviewer_o5oZ · 2023-10-30

**Soundness:** 2 fair
**Presentation:** 1 poor
**Contribution:** 3 good
**Rating:** 5
**Confidence:** 3

**Summary:**

The authors address the challenge of catastrophic forgetting in language models during continual refinement. They introduce a framework to predict which pretraining examples will be forgotten when a model is fine-tuned on new tasks. They propose some baselines and new approaches based on estimating logit change in pretraining sample when a model is finetuned on a new sample. Empirically, the proposed approach work only in one setting on BART models and fails on FLAN-T5 models (more commonly used and large scale).

**Strengths:**

- Problem statement is well motivated
- Thorough experimentation with proper baselines considered
- The paper builds up and formalizes an interesting problem which is going to be studied a lot in near future.

**Weaknesses:**

- Writing is quite poor for the methods section. Even after repeated reading, I could not understand how exactly $h$ is being learned for the trainable logit based forecasting. How does one train a LM which maps inputs to h(x), where h(x) is supposed to model the gradient of the model at x wrt $\theta$. I cannot find any details about this or some reference about this.
- Authors should clarify in Table 1 that they are predicting a minority class of forgotten examples in a binary classification setting. Hence F1 scores lower than 50% still make sense. In general, the writeup is quite poor.
- Poor empirical results on real large scale models : The proposed approach is heavily reliant on order 1 approximations of training dynamics, which do not hold true of large models as seen empirically. There are quite marginal gains of vanilla baseline of frequency based forgetting prediction on FLAN. Although I do thank the authors for acknowledging this fact, it still remains a major concern about efficacy and practicality of proposed approaches in the paper.

**Questions:**

See weakness section

---

> ### Author Response · Authors · 2023-11-20
> **Response to reviewer o5oZ**
>
> We thank the reviewer for thoughtful comments.
>
> **Q: Writing is quite poor for the methods section. Even after repeated reading, I could not understand how exactly h is being learned for the trainable logit based forecasting.**
>
> We thank the reviewer for bringing up the clarity issue. In the updated version, we added algorithm boxes for logit-based and representation-based forecasting methods to enhance clarity of the methods in Appendix C.
>
> Regarding the question about how $h(x)$ is computed: the logit-based forecasting method requires a kernel $\Theta$ that measures similarity between two examples. The first-order analysis in the beginning of Sec. 3.2 suggests that $\Theta$ should be dot products between gradients of the parameters, which is computationally infeasible, because it requires output length * vocab size backward passes to obtain all required gradients. We therefore propose to approximate with a lower-dimensional learnable kernel $\tilde{\Theta}=h(x)h(x)^T$. Here $h(\cdot)$ is a trainable LM encoder (either BART0 or FLAN-T5, depending on our experiment setup) and $h(x)$ has a much smaller dimension than actual gradients; $h(x)$ is explicitly trained to predict logit updates with a margin loss objective in Eq. 3.
>
>
> **Q: Authors should clarify in Table 1 that they are predicting a minority class of forgotten examples in a binary classification setting. Hence F1 scores lower than 50% still make sense.**
>
> We thank the reviewer for the suggestion. We have updated the captions in Table 1 to highlight the skewed nature of the class distribution.
>
> **Q: Poor empirical results on real large scale models**
>
> Thank you for pointing out the issue. We also acknowledged in abstract that logit-based forecasting performs well on BART0 but not on T5 models. However, for the other proposed method, representation-based forecasting, the performance improvement is clearer when we replay forecasted examples, as we see in Table 4. For example, we reduce EM drop ratio from 1.489 to 0.301 on FLAN-T5 with LoRA tuning.
>
> As a pioneering work on forecasting forgotten examples, we hope our findings can inspire future study on further algorithmic improvement.

---

> > ### Comment · Reviewer_o5oZ · 2023-11-21
> > **Significance of the numbers**
> >
> > I am still not convinced with the empirical results which authors provide. For example, in Table 4, the authors show a decrease in EM drop from 1-3% (with random replay) to almost no drop i.e. 0.5% with their approach. How significant this improvement is? It looks like the initial drop in EM itself on the $D_{PT]$ was low itself.
> > Alternatively put, how many more samples would have to be replayed with random replaying, if we wanted to achieve a similar EM ratio like the proposed approach? (I am sure that more random replay can match the performance, so it would be a good indicator to highlight the importance or significance of the numbers). I am fine even if the authors give a vague estimate.
> >
> > My concerns above also stem from Table 3, where the numbers do not make sense at all to me. The EM drop with vanilla FT seems to be < 1% (unless the numbers are in the range 0-1, which would be a weird choice). If the numbers are <1%, I would suggest the authors to choose a better problem setting where they show the efficacy of their approach.
> >
> > Finally, I take a strong exception to the authors self-claiming of their work as a "pioneering work" repeatedly, even in the response to the other reviewer. While I understand everyone loves their work (and no issues with that), please avoid these extravagant claims, as it is something one should leave to the community. I am not really sure which reviewer or org or conference gave this work the label of "pioneering work in forecasting forgotten examples"

---

> ### Author Response · Authors · 2023-11-22
> **Response to follow-up comments**
>
> We thank the reviewer for the prompt response and follow up comments.
>
> **Q: How many more samples would have to be replayed with random replaying, if we wanted to achieve a similar EM ratio like the proposed approach?**
>
> We thank the reviewer for suggesting this comparative study. We additionally show EM Drop ratio on FLAN-T5-large (Full FT) when we replay more mini-batches of upstream examples. In our default setup, we replayed 1 mini-batch every 10 update steps (Sec. 4.2), which corresponds to 3 mini-batches while learning a single online learning example (as we use 30 steps to learn an example).
>
> **Table: Fixing single errors (setups in Table 3)**
>
> #. Replayed examples | EM Drop Ratio %
> -- | --
> 3 | 0.068
> 6 | 0.064
> 15 | 0.122
> 30 | 0.138
>
> **Table: Continually fixing multiple errors (setups in Table 4)**
>
> #. Replayed examples | EM Drop Ratio %
> -- | --
> 3 | 1.129
> 6 | 0.869
> 15 | 0.038
> 30 | -0.141
>
> When fixing single errors, increasing the number of replayed examples over a certain limit causes increased forgetting. Replaying more examples cannot achieve improvement brought by Replay w/ Threshold (0.024) or Replay w/ Representation (-0.026) in Table 3.
>
> When continually fixing multiple errors, increasing the number of replayed examples consistently reduces forgetting. By comparing to the results in Table 4, the EM Drop Ratio % of replaying 3 mini-batches in Replay w/ Representation (0.582) is between that of replaying 6 to 15 mini-batches of random examples.
>
> In the manuscript, we included the results and discussions above in Appendix E.2.
>
> **Q: It looks like the initial drop in EM itself on $D_{PT}$ was low, especially in Table 3. I would suggest the authors to choose a better problem setting where they show the efficacy of their approach.**
>
> We agree with the reviewer that the forgetting of FLAN-T5 models is small when updating models on a single online learning example. In some problem setups, e.g., when online learning examples represent conflicting knowledge, we may expect more severe forgetting.
>
> However, our intention was to simulate a common setup where online learning examples and upstream data are not clearly conflicting. We see intriguing cases where learning one example causes forgetting of an irrelevant example (as exemplified in Figure 1).  For this purpose, we did not go over manual construction procedures of online learning examples; instead, we just collected mispredicted examples from some standard NLP benchmarks. As our results indicate, LMs like FLAN-T5 models only forget <1% of upstream examples (Table 3) (which also makes forecasting more challenging due to class imbalance). Still, we see the forgetting accumulates over continual updates and reaches 1-3% of EM Drop with replaying (Table 4) and may potentially get worse with a longer stream of online examples.
>
> After discussions with the reviewer, we consider it meaningful to extend to setups where forgetting is more severe in nature by (1) finding more dataset setups (2) longer data streams. Given the tight deadline, we have to leave them in future work. Still, we hope the results in the presented setup can be helpful to the community.
>
> Finally, it is not our intention to bring extravagant claims. I apologize for misunderstanding the degree of the term “pioneering work”. I just wanted to show the novelty of the problem setup. Thank you for pointing this out this language use issue.

---

> > ### Comment · Reviewer_o5oZ · 2023-11-22
> >
> > I thank the reviewers for their efforts in trying to resolve my concerns. As the authors themselves stated, this paper can gain much more from better empirical evaluations, where the comparisons (and the forgetting phenomenon to start with) are much more meaningful. I honestly feel that this paper can really gain a lot from another cycle of submission, where they try to incorporate my comments, search for better experimental settings and show better gains. Hence, I would like to maintain by current score of 5, "weak reject".

---

### Official Review · Reviewer_rXCe · 2023-11-03

**Soundness:** 3 good
**Presentation:** 3 good
**Contribution:** 3 good
**Rating:** 6
**Confidence:** 4

**Summary:**

In a continual learning framework focused on LM’s the authors study the problem of  predicting which samples from the upstream data a model is likely to forget after it has been trained on new data.  This is then also used to improve the samples selected for replay from upstream data (focusing on the ones to be forgotten).

**Strengths:**

- The authors focus on a timely and relevant setting of continual learning for LMs
- The approach is computationally efficient and reasonably well motivated
- The authors point out an interesting phenomenon of logit change
- Results of using forecasting for augmenting replay seem to be promising

**Weaknesses:**

- The experiments can benefit from some quantitative results about the computational efficiency, for example in Table 4 what is the overhead of the approach compared to replay w/random
- The authors describe several prior works on forecasting (albeit not in the LM space) it would be interesting to experimentally compare these methods to the proposals

**Questions:**

It would be interesting to know if the method and observations are applicable to classification problems or other continual learning domains

---

> ### Author Response · Authors · 2023-11-20
> **Response to reviewer rXCe**
>
> We thank the reviewer for thoughtful comments. We address the reviewer’s comments below.
>
> **Q: The experiments can benefit from some quantitative results about the computational efficiency, for example in Table 4, what is the overhead of the approach compared to replay w/random?**
>
> We thank the reviewer for the question. In the new version, we included the number of FLOP (floating point operations)  of forecasting methods in Appendix (assuming 100 examples per pretraining task and full fine-tuning of FLAN-T5-large) and added pointers in Sec. 5.4. We believe the FLOP are good empirical statistics that complement theoretical complexity analysis already presented in Sec. 5.4.
>
> Method | FLOP
> -- | --
> Representation-based | $1.35e^{10}$
> Trainable Logit-based | $2.15e^{11}$
>
> As a reference, without forecasting methods, obtaining ground truth forgotten examples requires   about $9.02e^{14}$ Flop.
>
> Furthermore, the number of replayed examples is controlled across all replay-based methods in Table 4. Therefore, the results above are the only overhead of replaying forecasted examples compared to replay w/random.
>
> **Q: The authors describe several prior works on forecasting (albeit not in the LM space) it would be interesting to experimentally compare these methods to the proposals**
>
> We discussed several related works that analyze features of forgotten examples and learning dynamics in Sec. 6. However, none of them studies the challenge of “forecasting”example forgetting. We believe we are the first to propose the task, set up baselines, and evaluate the performance of forecasting.
>
> **Q: It would be interesting to know if the method and observations are applicable to classification problems or other continual learning domains**
>
> We thank the reviewer for the suggestion. In this paper we formulated all NLP tasks as sequence-to-sequence generation tasks (Sec. 2) and experimented on P3, MMLU datasets that consist of diverse NLP tasks. These tasks include classification problems (e.g. sentient analysis, natural language inference) that are answered in a sequence-to-sequence format. We believe it is an interesting future work to apply the approach in other continual domains, such as vision domains, given the increasing interest in model refinement in various scenarios.

---

### Official Review · Reviewer_QYTw · 2023-11-07

**Soundness:** 2 fair
**Presentation:** 3 good
**Contribution:** 2 fair
**Rating:** 5
**Confidence:** 3

**Summary:**

This paper presents a method to predict what kind of information is forgotten when further training language models.
In particular, it proposes a partially inter-pretable forecasting model based on the observation that changes in pre-softmax logit scores of pre-training examples resemble that of online learning examples. Further a black-box classifier based on inner products has improved the forecasting performance. Based on the forecasting model, using examples that are forcasted to be forgotten to train the model can mitigate the forgetting issue.

**Strengths:**

1. The idea of predicting forgotten examples are novel.
2. The method is quite easy to understand and implement.
3. Solving the forgetting issue has practical application values in pretrained language models.

**Weaknesses:**

1. The experiment results reveal that the improvement over the baseline is quite marginal.
2. A more insightful experiments could be done to analyze why some examples are more import than others when training the model in terms of forgetting.
3. The relation between the further-training and forgetting is not quite clearly explained.

**Questions:**

1. Are the predicted samples sensitive to the training order of the same set of samples?
2. How the forgotten samples are related to the training data, learning rate, etc?

---

> ### Author Response · Authors · 2023-11-20
> **Response to reviewer QYTw**
>
> We thank the reviewer for the thoughtful comments. We address the questions and comments below.
>
> **Q: The experiment results reveal that the improvement over the baseline is quite marginal**
>
> Thank you for pointing this out. We also acknowledged in abstract that logit-based forecasting performs well on BART0 but not on T5 models.
>
> Nevertheless, the other algorithm we presented, representation-based forecasting, improves over baseline consistently. In fact, the performance improvement is clearer when we replay forecasted examples (Table 4). As an example, we reduce EM drop ratio from 1.489 to 0.301 on FLAN-T5 LoRA.
>
> As a pioneering work on forecasting forgotten examples, we also hope our findings can inspire further algorithmic improvement.
>
> **Q: Are the predicted samples sensitive to the training order of the same set of samples?**
>
> We perform additional experiments to address the question from the reviewer. We notice that examples that are forgotten at the end of the stream are not sensitive to the order of the examples. We see the standard deviation of F1 scores of the forecasting methods are small as we alternate the order of training examples (over 5 different orders).
>
> Method | Average(std) forecasting F1
> --- | ---
> Threshold | 28.91 $\pm$ 0.4
> Representation-based | 31.24 $\pm$ 0.9
>
> **Q: How are the forgotten samples related to the training data, learning rate, etc?**
>
> We thank the reviewer for the question. Certain factors, such as learning rate, have a clear effect on edit success rate and EM drop ratio (forgetting) in continual model updates, as shown below for FLAN-T5-large (full fine-tuning).
>
> Learning rate | Edit success | EM Drop Ratio.
> -- | -- | --
> 1e-4 |  95.7 | 24.897
> 1e-5 (default) |  95.7 | 3.302
> 2e-6 |  93.5 | 1.820
>
> When we evaluate the forecasting model trained in the default setup on other learning rate setups, we notice that the precision scores almost remain the same, indicating a common subset of examples are forgotten in three setups; while recall scores differ, because a great number of examples are forgotten only when using larger learning rates. We added the new results in the Appendix E.

---

> > ### Author Response · Authors · 2023-11-23
> > **Follow-up response to reviewer QYTw**
> >
> > We hope to extend our response to the reviewer, and we would appreciate any further feedback from the reviewer.
> >
> > **Q: More insightful experiments could be done to analyze why some examples are more important than others when training the model in terms of forgetting. The relation between the further-training and forgetting is not quite clearly explained.**
> >
> > We thank the reviewers for the comments. Our work is inspired by one of many possible definitions of interpretability: the degree to which a human can consistently predict the model’s result [1]. We believe our trainable logit-based forecasting method has made a step towards this definition of interpretability. The approach states that the logit updates on $x_{PT}$ upstream example depend on (1) the logit updates of the online learning example $x_R$ and (2) certain similarity $\Theta$ between two examples. Therefore, our study implies factors such as (1) $x_{R}$ with significant logit change on the label space of the upstream example $x_{PT}$ (2) $\Theta$ that is far from 0 may contribute to forgetting.
> >
> > We believe understanding and interpreting forgetting is a challenging problem that requires long-term effort. We will explore more factors that contribute to forgetting in future work.
> >
> >
> > [1] Molnar, C. Interpretable Machine Learning, 2022

---

### Author Response · Authors · 2023-11-20
**General response**

We thank all the reviewers for their thoughtful comments and suggestions. We appreciate the recognition of novelty and importance of studying the forecasting problem by the reviewers. We believe the problem will get increasingly relevant as the development of large LMs continues.

We have updated our manuscript to address valuable suggestions from the reviewers.
- We added algorithm boxes for training and inference procedures of forecasting methods in Appendix C and added pointers in our method section *[Reviewer o5oZ]*
- We added FLOP statistics in Appendix D to complement our computational complexity analysis in Sec. 5.4. *[Reviewers rXCe, PJD2]*
- We discussed effect of learning rates in model refinement and showed the generalization of trained forecasting models to different learning rates in Appendix E. *[Reviewer QYTw]*

We thank all the reviewers again for their time and valuable comments and we are happy to address further questions.

---

### Author Response · Authors · 2023-11-23
**Summary of the author-reviewer discussion phase**

We thank all the reviewers again for their review and feedback. We appreciate the positive follow-up comments by reviewer PJD2, and valuable follow-up suggestions (and the significant amount of time taken) by reviewer o5oZ.

To summarize the **strengths** recognized by the reviewers,
- Importance of the topic: forecasting forgotten examples has practical application and to be studied a lot in the future *(we thank all reviewers for recognizing the value of the study)*
- Novelty: The idea of predicting forgotten examples is novel *[QYTw, PJD2]*
- Efficiency: the proposed forecasting methods are simple *[QYTw, PJD2]* and efficient *[rXCE]*. In the updated version, we further presented FLOP statistics to complement computational complexity analysis in the initial version.

The main **weakness** pointed out by the reviewers QYTw and o5oZ is the small performance improvement, which has been the center of the discussion phase. At a high level, our evaluation consists of (1) performance (F1 scores) of forecasting methods (2) forgetting mitigation when replaying examples forecasted to be forgotten.

For (2), the marginal improvement mostly stems from the setup. Our intention was to study forgetting in a common setup of fixing one/sequence of errors of LMs in standard benchmarks. As the results indicate, the forgetting is not severe (which also made forecasting more challenging) on FLAN-T5 models. We agree with reviewer o5oZ that it is meaningful to extend the study to setups where forgetting is more significant, and **we will add them in the final version of the paper**. This includes natural setups, e.g. longer sequences of data, larger learning rates, and adversarial setups, e.g. when online learning data is noisy or conflicts with upstream data.

Nevertheless, we believe our presented setup is also representative of common practice, and we appreciate recognition by reviewers that the study under the current setup is presented with proper baselines [o5oZ] and thorough experiments [PJD2].

Here is a summary of updates in our latest manuscript.

- We added results of increasing the number of replayed examples and compared against replaying only a few examples forecasted to be forgotten in Appendix E2 *[Reviewer o5oZ]*
- We added EM scores of base LMs before updates in Appendix B.
- For other updates, please see our “General response” in earlier comments.

Thanks again to all the reviewers for their feedback and suggestions.

---

### Meta-Review · Area_Chair_LfNL · 2023-12-07

**Metareview:**

(a) Summarize the scientific claims and findings of the paper based on your own reading and characterizations from the reviewers.
- In the context of pre-trained LMs, the authors study the problem of forgetting past examples given updates. This is related to ideas explored in continual learning (including forgetting, replay, and gradient interference).
- The authors propose a few novel techniques for predicting which examples will be forgotten
- The empirical performance of the techniques varies across models

(b) What are the strengths of the paper?
- The methods proposed in the paper seem well-adapted to the problem (e.g, they likely scale better compared to prior works)
- The empirical analysis is precise and methodical. (the authors also report negative results)
- The paper introduces new ideas that seem promising based on empirical results

(c) What are the weaknesses of the paper? What might be missing in the submission?
- The paper could be better positioned within the continual learning literature. In particular, the authors cite the work of (Aljundi et al., 2019) and mention that this approach is computationally prohibitive in their setting. Providing more precise comparisons with this work (or others in continual learning that predict which examples will be forgotten) seems important and might even lead to interesting analyses. In particular, in the case of (Aljundi et al., 2019), wouldn't selecting samples from a randomly populated memory be a reasonable comparison?
- The authors also focus on forecasting accuracy. Forecasting seems most valuable as a diagnostic tool that can be used downstream (e.g., in a continual learning scenario to help with replay as you study in Table 3). Perhaps forecasting can be used to better understand properties of models, but then validations and comparisons with other techniques in the CL literature should be used.

**Justification For Why Not Higher Score:**

This paper has many elements of a very good paper, and, given the reviewers' assessments, it is in the borderline area.

As reviewers highlighted, the study reports marginal improvement in downstream evaluation. The authors are very clear about the quality of the results in the abstract, and given the relative novelty of the approach, I find these results promising.

However, I find that lack of comparisons to other baselines from the continual literature more problematic. Referring to a version of (Aljundi et al., 2019) as ground truth seems like an overstatement. I understand that the proposed work provides novelty in forecasting prediction, but again, as far as I understand, this is mostly a proxy that will be used downstream (e.g., to prevent forgetting in continual learning). I believe that comparing this type of work (Aljundi or, ideally, more recent work in this line of research) in terms of accuracy and computation would give a more precise understanding of the advantages of the current approach.

In Aljundi, the retrieval is done on a memory that contains only a subset of the whole training data. In that sense, those papers could be compared regarding forecasting accuracy and interpretability. The memory size likely controls compute-accuracy tradeoffs, which could provide comparisons with the proposed approach.

This was not an easy decision. As part of the process, I reached out personally to the reviewers and the senior area chairs. I am sorry that I cannot recommend acceptance at this stage, but I do believe there's a clear path to the next version of this work.

**Justification For Why Not Lower Score:**

N/A

---

### Decision · Program_Chairs · 2024-01-16

Reject